# Evaluation of Thin Wall Milling Ability Using Disc Cutters

**DOI:** 10.3390/mi14020341

**Published:** 2023-01-28

**Authors:** Adelina Hrițuc, Andrei Marius Mihalache, Oana Dodun, Laurențiu Slătineanu, Gheorghe Nagîț

**Affiliations:** Department of Machine Manufacturing Technology, “Gheorghe Asachi” Technical University of Iasi, 700050 Iasi, Romania

**Keywords:** thin wall, milling, disc cutter, residual stress, bending, machining deviation, aluminum alloy, influence factors, empirical mathematical model

## Abstract

In some cases, industrial practice requires the production of walls or parts with a thickness of less than one millimeter from a metal workpiece. Such parts or walls can be made by milling using disc cutters. This machining method can lead to the generation of residual stresses that determine the appearance of a form deviation characterized by bending the part or the thin wall. To evaluate the suitability of a metallic material for the manufacturing of thin walls by milling with disc cutters, different factors capable of exerting influence on the deviation generated by the residual deformation of the walls were taken into account. A test sample and an experimental research program were designed for the purpose of obtaining an empirical mathematical model. The empirical mathematical model highlights the magnitude of the influence exerted by different input factors on the disc cutter milling process regarding the size of the deviation from the form, and the correct position of the wall or thin part, in the case of a test sample workpiece made of an aluminum alloy. Input factors considered were cutting speed, feed rate, cutter thickness, wall or part thickness, thin wall length, and height. To rank the input factors whose increase leads to an increase in shape deviation, the values of the exponents attached to the factors in question in the empirical mathematical model of the power-type function were taken into account. It was found that the values of the exponents are in the order 0.782 > 0.319 > 0.169 for wall height, feed rate, and wall width, respectively. It was thus established that the strongest influence on the residual deformation of the thin wall is exerted by its height.

## 1. Introduction

Of the processes used in machining, those processes based on removing material from the workpiece by cutting have a special place. It can be seen that a large proportion of these processes are by cutting [1,2,3,4,5,6,7]. Essentially, machining is based on a strong compression of the workpiece material by the clearance surface of a cutting tool, until a shearing of the workpiece material occurs and material separates from the workpiece in the form of chips. The presence of a cutting process implies the existence of a sufficiently large force developed in the contact zone between the workpiece and the clearance surface of the cutting tool. The values of this force must ensure the conditions for overcoming the shear resistance of the workpiece material. Only then is it possible to generate chips. There are many machining methods (turning, milling, drilling, planing, mortising, and processes using abrasive particles, such as grinding, lapping, honing, polishing, etc.). Processes that use abrasive particles are often considered micro cutting, since at least some of the chip sizes are smaller than one millimeter.

As output parameters of a machining process, the material removal rate (assessed by the amount of material removed from the workpiece in a certain unit of time, for example in mm^3^/min), the accuracy of the machined part (dimensional accuracy, form accuracy, position accuracy, orientation accuracy, tapping accuracy), the roughness of the machined surfaces (currently, there is a relatively large number of parameters that can be used to evaluate the roughness of the surfaces), the thickness of the layer affected by machining, the presence and nature of residual stresses in the surface layer, etc. [8,9], can be taken into account. Such output parameters are also used in the case of micro-machining processes, considering that the dimensions characterizing the machined surfaces must be, for example, smaller than 1 mm.

Since machining processes in general, and micro-machining processes in particular, are based on the generation of cutting forces, some errors can be expected when it is necessary to obtain thin walls by detaching material by machining of thin parts. It is thus possible for residual deformations to appear, which constitute deviations from the form of the machined surfaces (strictly flat surfaces are no longer obtained) or from the orientation of the flat surfaces generated by machining (for example, the appearance of deviations from the perpendicularity of the machined surfaces to other surfaces). There is concern among researchers about better understanding the processes that lead to a residual deformation of thin walls in the case of some machining processes.

Thus, in the first decade of the current millennium, groups of researchers coordinated by Ratchev published the results of research on modeling the behavior of low-rigidity components produced by machining, highlighting the factors able to exert influence on this behavior and the possible ways to reduce the errors that appear as a result of the machining [10,11,12,13,14].

Monitoring the variation of some output parameters of the cutting process and ensuring the possibility of intervention before the processing errors reach unacceptable values could be a solution for increasing the processing precision of thin walls. In this sense, in works developed by Guo and his collaborators, the problem of monitoring the machining processes of thin components of complex parts was addressed [15,16,17]. The difficulties of such monitoring were highlighted, and the design criteria of usable monitoring sensors were proposed.

The influence of some machining conditions on the surface dimensional error generated when using peripheral milling of the thin-walled workpiece was highlighted by Wan et al. [18]. They proposed the use of optimization methods to ensure high productivity without sacrificing machining accuracy. An interesting aspect addressed by the paper’s authors was considering an existing error compensation method.

The possibilities of using finite element analysis to model distortion or deflection of the thin-walled parts generated by milling with a helical endmill were examined by Izamshah et al. [19]. They proposed a machining model capable of providing information on the generation of errors in the mentioned category when using an end milling process.

In a book chapter published in 2012, Huang et al. considered that temperature and forced-induced deflection could generate surface errors that could only be made through a milling process [20]. The coupled effect of the two factors was analyzed and a robust spindle speed optimization was developed.

In a book chapter devoted to the presentation of applications of finite element analysis in mechanical engineering, Huang et al. [21] examined thin-wall deflection due to the thermomechanical effects specific to the cutting process for this purpose, using finite element analysis. They aimed to analyze how the uncertainties specific to the milling process influence the stability of the process and the location of surface errors. The objective pursued was the optimization of the milling speed.

Some possibilities for making thin-walled parts using additive manufacturing technologies were analyzed by Isaev et al. [22]. The manufacture of thin-walled parts from thermally treated titanium-based powder materials by electron beam melting was considered. Milling was then used to finish the surfaces made by electron beam melting. It was found that form accuracy and surface roughness are influenced by variable wall stiffness, which could cause the appearance of vibratory processes, with negative effects on the quality of thin walls. The use of additive manufacturing methods has also been a concern for other researchers [23,24,25,26].

An investigation of the mechanism that generates thin-wall deformation due to the use of a milling process in the case of an aluminum alloy workpiece was undertaken by Wu et al. [27]. It was observed that using a quasi-symmetric machining method, the value of the maximum deformation can be reduced by up to 20% of the value of the deformation generated by the application of the traditional one-side machining method.

A manufacturing solution of thin walls can be based on the use of sacrificial support structures in the case of a hybrid manufacturing process [28]. Afterward, the sacrificial support structures are removed. One of the conclusions of the research undertaken by Vaughan et al. highlights the influence of some geometric characteristics of sacrificial support structures on thin wall geometric accuracy as a result of the application of a down milling process.

Del Sol et al. developed a synthesis of research results related to the machining of thin-walled light alloy parts [29]. They proved the growing interest in solving such a problem by the almost continuous increase in the number of scientific works published between the years 2000 and 2018 on the subject. The identification of some solutions for improving the stability of the machining process and reducing deformations in the case of generation by cutting thin walls was highlighted.

An extensive analysis of geometric errors generated in thin-wall machining processes was performed by Wu et al. [30]. Aspects related to thin-wall deformation were analyzed from the point of view of theories and mechanisms specific to machining processes. The possibilities of predicting, monitoring, and eliminating the deformations of thin walls as a result of the machining processes were highlighted.

To diminish the deformation of the thin-walled workpiece capable of significantly affecting the machining accuracy and surface quality, Ma et al. [31] considered a dynamic characteristic reconfiguration of a fixture-workpiece system, mainly aiming at reducing the intensity of vibrations specific to the milling process.

An attempt to analyze the problems related to the generation of thin parts or components of thin-walled parts by machining, as different researchers have approached them, highlights the fact that the main objectives pursued were the following:-The identification or generation of software capable of analyzing the behavior of the workpiece material when the problem of generating thin walls by chipping arises [32,33];-The influence of the rigidity of the technological system or some of its components on the possibilities of generating thin walls by chipping [10,11,12,13,34,35,36,37,38];-Highlighting the connections between the vibrational or dynamic processes in general, specific to the realization of thin walls by cutting, and some aspects of the characterization of the surfaces generated in this way [39,40,41,42,43,44,45,46,47,48,49,50,51,52];-The use of the finite element method for modeling some aspects related to the generation by cutting thin walls [31,33];-The evaluation of the size of the cutting forces that occur when thin walls are obtained by cutting and revealing the correlation between the cutting forces and the deformation of the thin walls [2,12,35,53,54,55];-The identification of solutions that lead to the reduction of the deformation of thin walls generated by cutting [21,28,56,57,58];-Addressing the problem of deformations generated by manufacturing some parts by cutting or by 3D printing [13,22,27,28,39,56,57,58,59];-Optimization of the machining process of thin-walled parts [18,60,61].

In this paper, some research results are presented, with the intention of generating empirical mathematical models to highlight the influence of some input factors on the machining process of a size that characterizes the error generated when obtaining thin walls.

A machining process using disc cutters to generate thin walls was considered. Consideration was given to the design of a test sample that would allow the measurement of multiple values of the errors generated by a single working stroke of the disc cutter.

## 2. Materials and Methods

### 2.1. Background

Suppose it is necessary to machine a groove in a parallelepiped-form workpiece (Figure 1a) by milling with a disk cutter, resulting in a thin wall (Figure 1b). For smaller thicknesses *t* of the thin wall, the mode of action of the milling teeth and the forces generated during milling could lead to a residual deformation of the thin wall generated by milling, as seen in Figure 1c.

A cutting scheme corresponding to the groove and thin wall generation mode can be seen in Figure 2. The disc cutter will rotate with speed *n_T_*, while the workpiece will make a feed movement *f_w_*. So-called feed milling is preferred, resulting in a common direction of the vectors corresponding to the cutting speed and feed speed in the area of contact between the disc cutter and the workpiece material.

Considering an *xyz*-coordinate system attached to the tool at the level of the contact area of the cutting tooth with the workpiece material (Figure 3), the total cutting force is decomposed into three components, *F_x_*, *F_y_*, and *F_z_*, as can be seen in Figure 3. In this case, the *F_z_* component of the cutting force will tend to deform the thin wall towards the outside of the workpiece. If the thin wall is nevertheless sufficiently thick, and has, therefore, a high stiffness, and the *F_z_* component has a small size, only an elastic deformation will develop that will not manifest in a deviation of the wall from its correct form (Figure 1b). 

However, if the previous conditions are not met, the pressure generated by the *F_z_* component of the cutting force may be greater than the strength of the workpiece material, and the thin wall will buckle. This curvature is, in fact, a result of the action of several cutter teeth that gradually contribute to the generation of the groove that separates the thin wall from the workpiece. The consequence of such processes will be a smaller or larger curvature of the wall (Figure 1c).

### 2.2. Analysis of the Cutting Process with Disc Cutters

From the point of view of machining accuracy, the curvature (bending) of the thin wall is a machining deviation, and it is important to avoid the conditions that could generate that deviation.

Achieving thin walls by milling may be possible using distinct tools such as disc cutters, end mills, and barrel-shaped cutters [12,54]. The simplest method is the one that uses disc cutters. A worn disc cutter can be resharpened with relative ease. Barrel-shaped cutters could be of particular interest when the bottom of the channel that separates a thin wall from the rest of the workpiece material is required to present a curved shape in cross-section, to ensure, for example, a better behavior of the part under stress fatigue. However, barrel-shaped cutters are more expensive, more difficult to resharpen, and will generate a different material behavior during the thin wall generation process.

End mills could be used for large groove widths that separate the thin wall from the rest of the workpiece material. Their use could ensure higher productivity of thin wall generation, but the cutting process has many different aspects than when using disc cutters.

Aiming to identify a way to evaluate the capacity of a material to obtain thin walls through disc cutter milling, it was considered possible to use a systemic analysis to highlight the input factors in the milling process that could affect the respective capacity of the investigated material.

In principle, the systemic analysis of a process, phenomenon, or object is based on the transfer of the characteristics of a system to the entity under consideration. This implies the existence of input factors, output parameters, and disturbance factors, respectively. It can note that the systemic analysis also aims to identify the possible correlations between the input factors and the system’s output parameters, and the interactions between the input factors in the system.

The analysis revealed the following groups of input factors in the process under investigation:(a)The physical-mechanical properties of the workpiece material. A higher mechanical resistance could lead to lower values of the deviation ε;(b)The dimensions that define the workpiece and the groove to separate the thin wall (height *h_w_* and width *w_w_* of the thin wall, width *w_c_* of the groove (in direct correspondence with the thickness *t_t_* of the disc cutter) (Figure 4);(c)The number of cutting teeth and the dimensions that define the geometry of the cutting teeth of the disc cutter (angles, edge rounding radii, etc.). An important influence on the size of the shape deviation of the thin wall can be exerted by the geometric characteristics (back rake angle, tool nose radius, side clearance angle, etc.) of the secondary cutting edge of the disk cutter that generates the thin wall. Such characteristics determine the magnitude of the *F_z_* component of the cutting force that will lead to the deformation of the thin wall;(d)The level of wear of the active edges of the cutting teeth of the disc cutter;(e)The values of some parameters characterizing the machining conditions (cutting speed *v_c_*, feed rate *f*);(f)The physical-mechanical properties of the disc cutter material. A possible elastic deformation of the cutting teeth along different directions could lead to the modification of the deviation value *ε*.

Given the characteristics of the milling process used to generate thin walls, it is expected that the machined surface roughness of the thin wall will also be significantly affected by the cutting conditions [5,29,55,62,63,64]. The successive entry of the cutting teeth of the tool into the material of the workpiece, the variation of the chip thickness, the variation of the cutting force, the vibrations generated by the milling process, the values of the milling parameters, etc., can be factors with a strong influence on the values of some parameters characterizing the roughness of the machined surface of the thin wall.

In the systemic analysis, those input factors in the investigated process whose values cannot be adjusted by the operator but whose variation could affect the output parameter values are considered disturbing factors. Thus, the uncontrolled variation of some of the input factors already mentioned could be considered. For example, variations in the mechanical properties throughout the test sample material, the values of the machining parameters, or the values of some geometric characteristics of the cutting zone of the tool could affect the size of the deviation *ε* from the correct form of the thin wall, sometimes significantly.

In the case investigated in this paper, the main output parameter of the investigated process is the deviation from the correct form of the thin wall. It was agreed to use the displacement *ε* of an edge corresponding to the free end of the thin wall to the position of this edge before milling with a disk cutter as the evaluation size of the permanent deformation of the thin wall.

The existence of a large number of input factors in the processing of thin walls with disc cutters, and the large ranges of variation of the values of these factors, could generate difficulties in optimizing such processes [65]. Using automatic learning techniques could facilitate the processing of experimental results, and therefore the identification of optimal conditions for obtaining thin walls.

One output parameter of the thin-wall machining process with disc cutters could be the size of the burrs generated by chipping. The removal of burrs will require the use of deburring processes after the end of the milling process. For this reason, introducing deburring methods involving the help of robots could prove an effective solution for reducing the duration of manufacturing products incorporating thin walls [66].

The elasticity of the workpiece material can also influence the size of the shape deviation generated when machining with disc cutters. Suppose the magnitude of the *F_z_* component does not generate stresses greater than the elastic strength limit of the test sample material. In that case, an elastic recovery of the thin wall to the correct desired shape will occur after the passage of the disc cutter. However, since the mechanical stress generated by the disc cutter usually exceeds the elastic strength limit of the workpiece material, it is expected that the thin wall will be affected by the appearance of a shape deviation.

### 2.3. Finite Element Modeling Elements

For a better understanding of the processes that develop when obtaining flat surfaces with the help of disc cutters, a finite element method was used, allowing the identification of some detailed information about what happens in the cutting area. Some results of using the finite element method (FEM), in this case, can be seen in Figure 5.

Using FEM, a 3D assembly consisting of a cutter and a workpiece was designed. It was assumed that a groove would be machined in one step. Both components were individually meshed, taking into account the teeth of the tool that come into direct contact with the workpiece. The mesh uses a hex dominant method for the cutter geometry, with quadratic elements as the preferred type. Due to the cutter’s geometry, some regions are meshed with triangles instead of quads. Both bodies are sized, the cutter with an element size of 3 mm and the workpiece with an element size of 0.63 mm. A face sizing method with a 2.5 mm element size was used for tooth faces that enter into contact with the blank in the simulation. Both meshed bodies can be observed in Figure 5a.

Thus, a process of removing material from the workpiece was simulated. As initial conditions, linear and angular velocities were assigned to the cutter, which received values along the X and Z axes for the velocity condition and on the Y component of the global coordinate system for the angular velocity expressed in rad/s. The bottom surface of the workpiece was fixed. Appropriate materials were assigned to both the cutter and workpiece. The material for the blank is based on Al7039, found in the Explicit Materials section of the Ansys library. The cutter received a structural steel material since we aimed for results concerning the workpiece.

FEM was applied to obtain customized results on process energy values that can otherwise be very difficult to measure. User-defined results of the form INT_ENERGYALL were generated. These results provide an image of the energy stored in both the tool and the workpiece due to the deformation processes [67]. Assuming an energy probe is used, it is noticed that the energy reached −707.52 J for the minimum value over time at the contact between the cutter and the workpiece, taking into account that the tool travels the full length of the workpiece. A maximum value over time at contact of 0 J was thus reached. From the graphic representation (Figure 5), which shows the internal energy distribution recorded for both geometries, we can see the thickness of the area affected by the milling process, and where residual stresses are generated that lead to the plastic deformation of the thin wall.

The graphic representation also highlights the way the low walls are formed, as the cutter makes the feed movement. The analysis also revealed the existence of a reduced recovery of the workpiece material in the area of the newly formed walls of the groove, which leads, in this case, to a shape of the generated groove close to that of the actual test samples. In the current setup, we may notice in an exaggerated manner in the way chips are formed and removed, as well as burrs that remain after the cutting process is complete. The thin wall itself suffers as the cutter passes by, moving sideways almost in waves, corresponding to each tooth action. The simulation reveals the dependency between the degree of surface finish and the number of teeth as well as the rotational speed of the cutter. The current setup uses a lower number of teeth, especially to highlight the areas affected by the cutting edge and those that are following.

The authors acknowledge that results need further refinement. A finer mesh could be used, but that would require much greater computational power than that available to the authors. A tool with exact number of teeth may also be considered.

### 2.4. Experimental Conditions

As a metallic material for the test samples used in the experimental tests, a less hard metallic alloy was considered, which would not generate intense wear of the disc cutter during the experiments, as such wear would have occurred as an additional disturbing factor in the determination of an empirical mathematical model of the deviation from the correct form of the thin wall. For this reason, it was preferred to use an aluminum alloy (which contained 95.9% aluminum), from which the test sample used in the experimental tests was made.

Later, it was necessary to establish the factors that would be taken into account in the experimental research, from among the factors revealed by the systemic analysis of the capacity of a material to ensure conditions for obtaining thin walls by cutting. The following input factors (independent variables) were thus established in the investigation process: the dimensions that characterize the wall (thickness *t_w_*, width *w_w_*, height *h_w_*), some of the cutting parameters (feed rate *f*, cutting speed *v_c_*), the thickness *t_t_* of the disc cutter, if such a cutting tool is used to generate a thin wall.

It was assumed that, for the considered domains of variation of the input factors, there would be a monotonous variation of the deviation from the correct position of the thin wall generated by machining with the disc cutter. This situation would limit the experimental trials to be carried out only at two levels of variation of the considered process input factors.

Also, to reduce the time required for the experimental tests, it was proposed to use several experimental tests following the requirements of an L8-type Taguchi array, with seven input factors (independent variables) at two levels of variation. It was later found that it was possible to use only six input factors instead of the seven factors originally considered. Since the software used to generate an empirical mathematical model [37] is based on the use of the least squares method, it was noted that reducing the number of input factors, compared to that corresponding to an L8-type Taguchi array, would not generate difficulties, since the number of experimental tests remained higher than the number of input factors (of independent variables).

The actual values of the input factors were entered in the first columns of Table 1. As mentioned, two levels of variation, maximum and minimum, of each of the input factors considered were established. Initially, two levels of variation of the feed rate *f*, cutting speed *v*, the thickness *t_t_* of the disc cutter, the width *w_w_* and the height *h_w_* of the thin wall, and respectively the thickness *t_w_* of the thin wall were established.

The experimental research was carried out on an FUS 25 universal tooling milling machine (manufactured by Mechanical Enterprise Cugir, Cugir, Romania). As cutting tools, high-speed steel disc cutters with a diameter of 65 mm, and a thickness of 1.2 mm and 1.54 mm, respectively, were used. A digital caliper was used to determine the deviations from the planar form of the thin-walled surfaces generated by milling. To ensure better stability of the milling process, the method was preferred so-called counter-feed milling.

For the experimental evaluation of the capacity of a material to allow the generation of thin walls by machining with a disc cutter, a test sample was also designed, to minimize the time needed to perform the experimental tests. As seen in Figure 6, this test sample shows several columns, along which areas were created to present thin walls of different lengths and heights, by previously milling some grooves.

In principle, the values of the cutting parameters were established, taking into account the indications from the specialized guidelines and the actual possibilities offered by the machine tool on which the experimental tests were intended to be carried out. Thus, the recommended values of the feed for a tooth of the disc cutter were considered, but taking into account the number of teeth of the disc cutter and the rotational speed of the disc cutter, the feed rate values entered in Table 1 were calculated. In the case of the width *w_w_* of the thin wall, however, three values were used, so that these values could be used in the mathematical processing of the experimental results to generate the empirical mathematical model highlighting the influence exerted by some process input factors on the size *ε* of the deviation from the correct form of the thin wall.

As thicknesses of the disc cutters with which to perform the experimental tests, values *t_t min_* = 1.20 mm and *t_t max_* = 2 mm were initially used. It was found, however, that in the case of a thickness of the disc cutter *t_t max_* = 2 mm, some thin walls tilted significantly, touching either the horizontal surface of the bottom of the groove, or the surface of the other wall of the groove, in the test sample (an example is the one shown in Figure 7b). For this reason, another disc cutter was considered, with a thickness of *t_t max_* = 1.54 mm.

Two images of some test samples intended to highlight how the deformation of the thin walls occurred as a result of machining the grooves using disc cutters can be seen in Figure 7. 

## 3. Results

The results obtained by measuring the deviation ε from the planar form of the surface were also entered in Table 1.

The determination of an empirical mathematical model was aimed at illustrating how the values of some factors characterizing the machining conditions exert influence on the size of the deviation ε. The experimental results were mathematically processed using specialized software based on the use of the least squares method [38]. The value of Gauss’s criterion was used as a parameter to evaluate the adequacy of the mathematical model to the experimental results. This value is calculated as a ratio between the sum of the squares of the differences between the values of the ordinates corresponding to the use of the proposed mathematical model and, respectively, the values of the ordinates related to the experimental results, the sum of the squares being divided by the difference between the sum of the experimental test number and the number of constants in the dependence relationship [68,69,70]. The lower the value of Gauss’s criterion, the more the mathematical model is appropriate to the experimental results.

The empirical mathematical model determined using the specialized software, in the case in which the lowest value of Gauss’s criterion was reached among the five models considered, is of the form of a polynomial of the second degree:*ε* = −2.042 + 0.0397*v_c_* − 0.000207*v_c_*^2^ − 633.851*f_z_* + 83447.07*f_z_*^2^ + 6.593*t_t_* − 4.193*t_t_*^2^ + 14.373*t_w_* − 13.568*t_w_*^2^ + −0.887*w_w_* + 0.0662*w*_w_^2^ − 0.361*h_w_* + 0.00108*h_w_*^2^,(1)
the value of Gauss’ criterion being *S_G_* = 3.989648.

On the other hand, it was found that for the modeling of specific aspects of such machining, the use of mathematical power-type functions is frequently used. The main advantage resulting from using power-type functions results from directly obtaining information regarding the intensity and direction of the action of an influencing factor by simply examining the exponents attached to each influencing factor in the empirical mathematical model. However, there is also a limitation in the field of using empirical mathematical models of the power-type function in situations where the hypothesis is accepted that there will be a monotonous variation (without maxima and minima) of the output parameter to the variation of the input factors. Under the previously mentioned conditions and accepting a monotonous variation of the input factors, the empirical mathematical model of the power-type function determined using the specialized software is, in this case, of the form:*ε* = 3.393*v_c_*^−0.0534^*f_z_*^0.319^*t_t_*^−0.347^*t_w_*^0.0332^*w_w_*^0.169^*h_w_*^0.782^,(2)
for which Gauss’s criterion has the value *S_G_* = 4.388665.

The empirical mathematical power-type function model was used to develop the graphical representations in Figure 8 and Figure 9.

The analysis of the empirical mathematical model defined by Equation (2) shows that the strongest influence on the deformation *ε* of the thin wall is exerted by the height *h_w_* of the wall, since the exponent attached to this quantity has the highest absolute value, compared with the values of the other exponents. Such a result was expected, considering that increasing the wall height means an increase in the cantilever length of a component of the test sample, which leads to an increase in the bending under the action of some components of the force generated by the cutting process when using a disc cutter. An important influence on the size *ε* of the residual deformation of the thin wall is still exerted by the the feed rate *f_z_*. With the increase in the value of feed rate *f_z_*, there is an increase in the component of the cutting force acting on the thin wall, and in this way the increase in the value of the deformation of the thin wall can be explained.

It is found that a small decrease in the residual deformation of the thin wall occurs when the cutting speed *v_c_* is increased, since the exponent attached to this factor in Equation (1) has a low negative value. When the cutting speed *v_c_* increases, there is a decrease in the value of the component of the cutting force that causes the deformation of the thin wall due to a better transformation of the material of the workpiece into chips, and this could result in the reduction of the deformation *ε* of the thin wall. The improvement of the flow conditions of the test sample material when it is transformed into chips can also be a consequence of the improvement of the plasticity of the material as a result of the higher temperatures generated by the heat increase. It is known that an increase in cutting speed leads to an increase in temperature in the area corresponding to the tips of the teeth, which improves the plasticity of the aluminum alloy from which the test sample was made.

On the other hand, the milling process is characterized by a variation of the cutting forces determined by the variation of the chip thickness at the entry into the cutting process of each of the teeth of the disk cutter. This fact also causes a variation in the *F_z_* component of the cutting force (Figure 3). However, given the large number of teeth of the disc cutters used in the experimental research, it is to be expected that the variation in the size of the component *F_z_* of the cutting force occurs in a relatively small range of the values of the component *F_z_* of the cutting force. For this reason, it can be considered that the deformation of the thin wall of the test sample will be less affected by the variation in the cutting process specific to the use of the disc cutter, and an average value of the magnitude of the force component *F_z_* could be considered.

The wear of the cutting tool could also exert an influence on the size of the cutting force generated in the machining process with the disc cutter. However, since the aluminum alloy has relatively low mechanical strength and hardness, it is expected that the disc cutter (made of hardened steel) will not experience significant (at least, noticeable with ordinary measuring tools) wear during its use in the experimental research. For this reason, no considerations were made regarding the influence of cutting tool wear on the permanent deformation of the thin wall.

Among the three parameters that define the dimensions of the thin wall (thickness *t_w_*, width *w_w,_* and height *h_w_*), as already mentioned before, the strongest influence is exerted by the height *h_w_* of the thin wall. This height is analogous to the length in the cantilever of a bar subjected to bending. On the other hand, among the dimensional characteristics of the thin wall, the wall thickness *t_w_* seems to exert the least influence, and even a negligible influence, since the exponent attached to this parameter in Equation (2) has a very low value.

Less expected is the influence exerted by the milling cutter thickness *t_t_*, according to the empirical mathematical model constituted by Equation (2). From this equation, it can be seen that an increase in the milling cutter thickness *t_t_* causes a reduction in the residual deformation *ε* of the thin wall. For the conditions under which the experimental tests took place, this could be explained by a reduction in the magnitude of the component *F_x_* of the cutting force that causes the deformation of the thin wall in relation to the total magnitude of the cutting force generated by the cutter disk.

It can be underlined that some of the observations attributed to the influence of different input factors on the residual deformation *ε* of the thin wall could also be explained by the deficiencies of the empirical mathematical model of the power-type function. This mathematical model cannot highlight the existence of maxima or minima of the function under consideration. As can be seen, Equation (1), corresponding to a higher extent to the experimental results, highlights the existence of a maximum or minimum of the size *ε* of the residual deformation of the thin wall. (If the value of Gauss’s criterion is analyzed, it is found that the function defined by Equation (2) leads to a lower value of Gauss’s criterion, this being, in fact, the most appropriate empirical mathematical model, among the five functions analyzed by the software used, namely polygon of the first degree or second degree, power-type function, exponential function and hyperbolic function). 

The increase in the residual stresses (stresses that determine the values of deviations *ε* from the planar form of the thin wall) when increasing the feed rate value is consistent with those mentioned by Jiang et al. [59]. The fact that a decrease in cutting speed can lead to a reduction in residual stresses capable of generating a deformation of the thin wall was also reported by Jiang et al. [59].

## 4. Conclusions

Using different processing methods to manufacture thin parts, or areas of a part that constitute thin walls, is a problem of current interest, as can be stated by considering the results obtained by other researchers to solve such a problem. These results confirm that, under certain conditions, cutting processes can generate sufficiently high residual stresses such that the thin wall deforms, i.e., a deviation from the desired planar wall form occurs. To obtain more detailed information regarding the conditions for the appearance of the deviation from the planar form of the thin wall, experimental research was designed, and different machining conditions with disc cutters were used. To reduce the time required for the experimental tests, a specially designed test sample was used, so that thin walls with different dimensions could be obtained with a single stroke of the disc cutter. The mathematical processing of the experimental results allowed the identification of some empirical mathematical models capable of highlighting the influence exerted by the cutting speed, feed rate, thickness of the disc cutter, and the width, height, and thickness of the thin wall generated by milling on the value of the deviation from the flat form of the thin wall. It was found that the strongest influence is exerted by the height of the thin wall, the increase of which can lead to a significant increase in the deviation from the flat form. Another factor whose increase causes an increase in the deviation from the flat form is the feed rate. The order of influence of the first three factors entering the process, and whose increase leads to the increase of the considered shape deviation, is highlighted by the values of the exponents attached to each of the factors in the empirical mathematical model of the power function type. The values of the respective exponents are in the order 0.782 > 0.319 > 0.169 for wall height, feed rate, and wall width, respectively. In the future, it is intended to continue the research on obtaining thin walls through other machining procedures and using other materials for samples.

## Figures and Tables

**Figure 1 micromachines-14-00341-f001:**
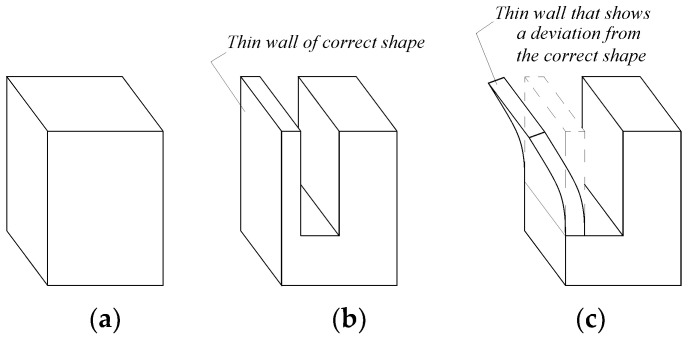
Bending of the thin wall obtained by milling with a disc cutter: (**a**) the initial shape of the semi-finished product; (**b**) the desired shape of the thin pre-net; (**c**) the shape of the thin wall generated by milling with a disc cutter.

**Figure 2 micromachines-14-00341-f002:**
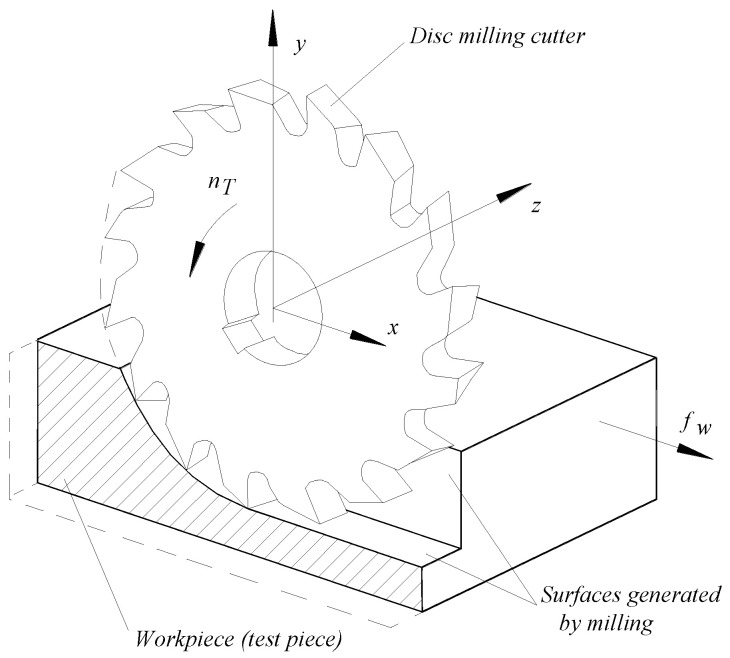
Cutting scheme when milling with a disc cutter.

**Figure 3 micromachines-14-00341-f003:**
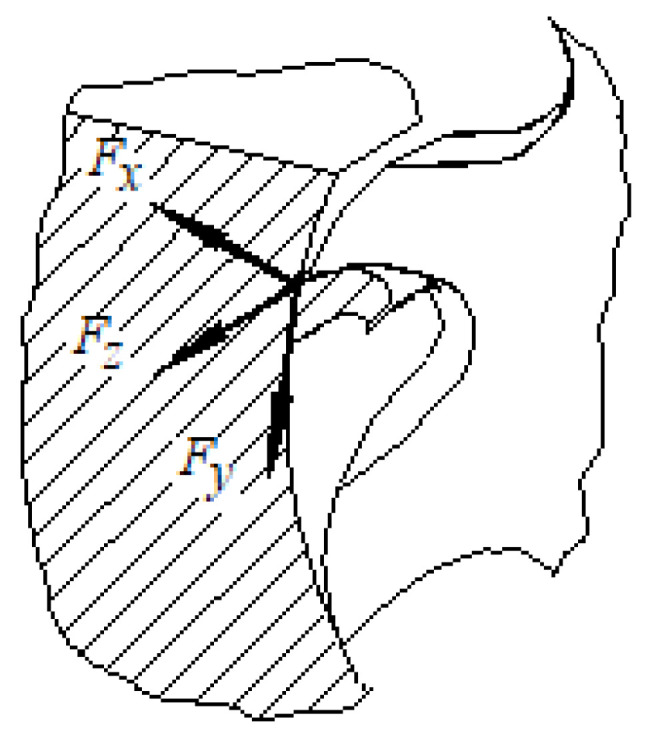
Decomposition of the cutting force with which the peak of a cutting tooth of a disk cutter acts on the stock material.

**Figure 4 micromachines-14-00341-f004:**
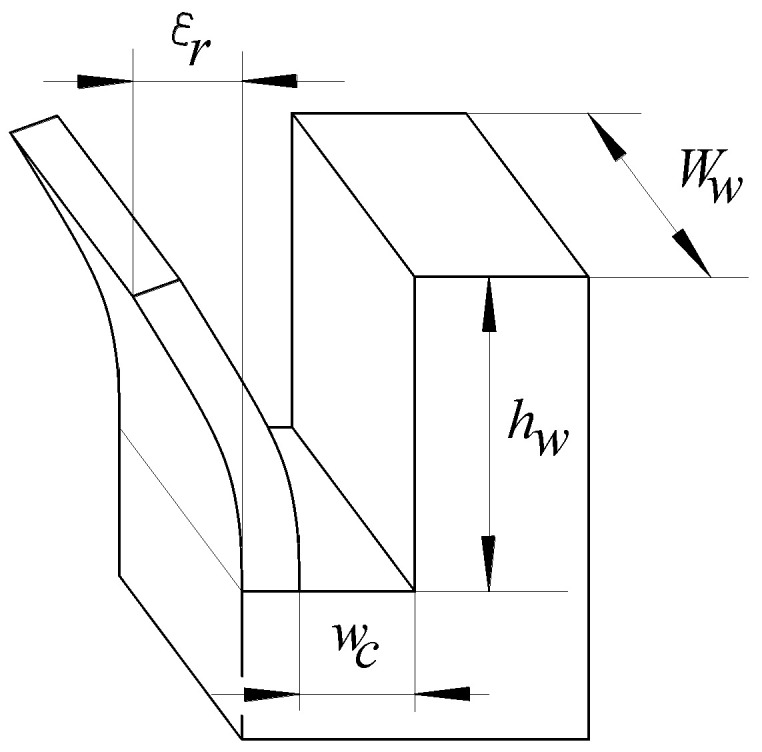
Dimensions that characterize the thin wall’s bending when machining with a disc cutter.

**Figure 5 micromachines-14-00341-f005:**
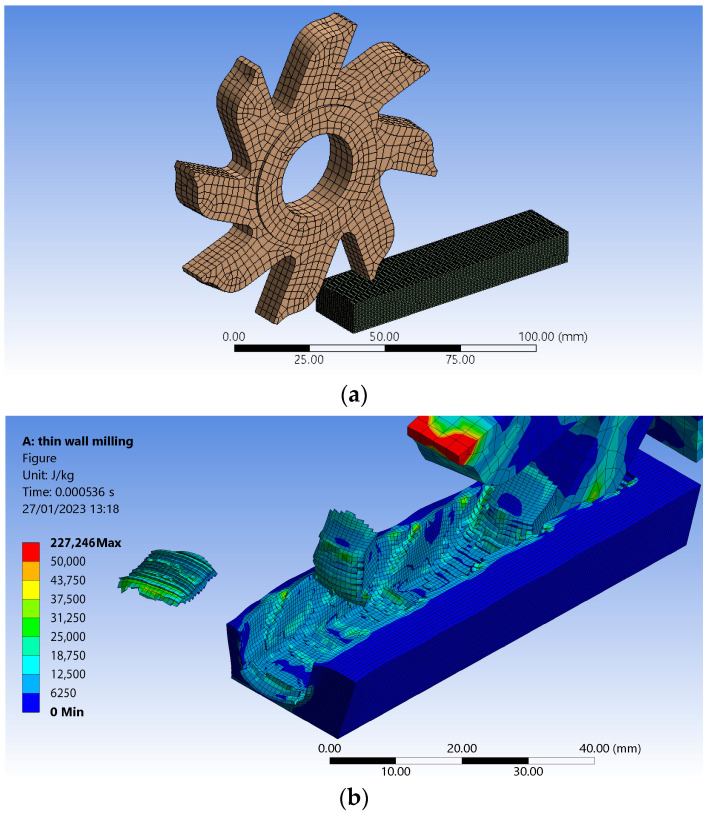
Highlighting using the finite element method: (**a**) the mesh specific to the disc cutter and the workpiece; (**b**) the energy transferred to the surface layer generated by milling with a disc cutter.

**Figure 6 micromachines-14-00341-f006:**
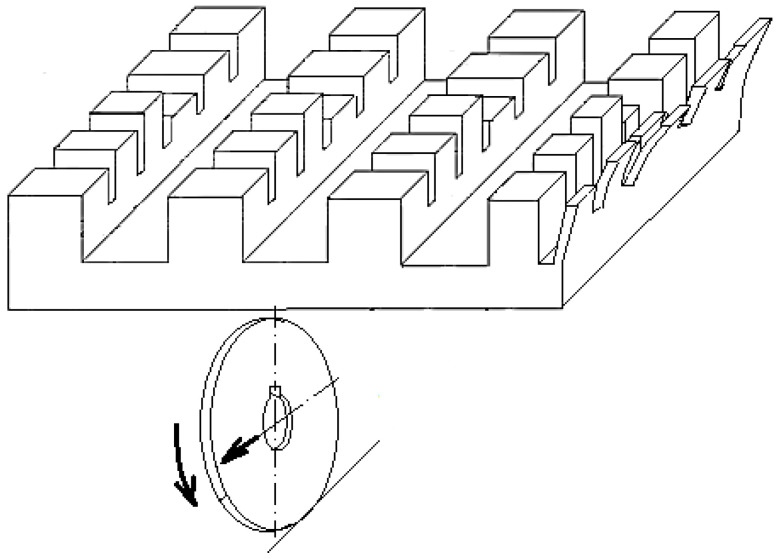
The shape of the test sample proposed for performing the experimental tests after the first stroke of the disc cutter.

**Figure 7 micromachines-14-00341-f007:**
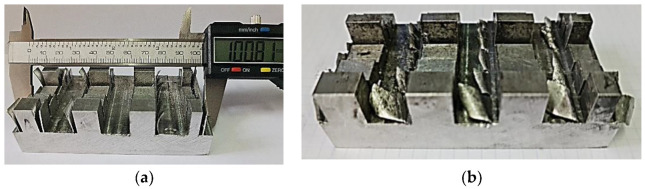
Images of the results obtained for different thicknesses of the disc cutter (**a**) *t_t min_* = 1.2 mm; (**b**) *t_t max_* = 2 mm; the deviation from the correct form of the thin wall was so high that the curvature of the wall led to touching the bottom of the groove or the other wall of the groove.

**Figure 8 micromachines-14-00341-f008:**
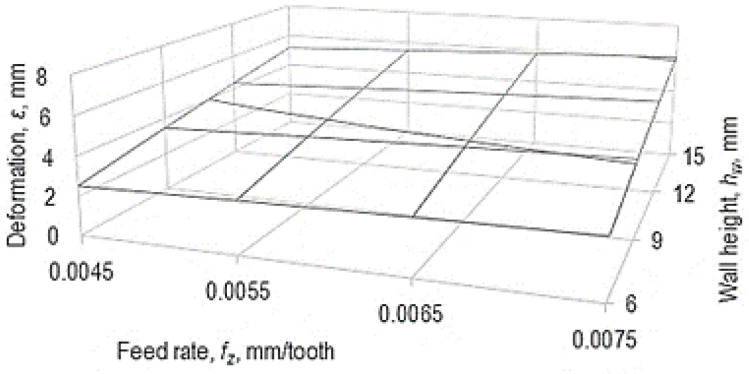
Influence exerted by the feed rate *f_z_* and the wall height *h_w_* on the value of deformation *ε* (*v* = 100 m/min, *t_t_* = 1.2 mm, *t_w_* = 0.2 mm, *w_w_* = 10 mm).

**Figure 9 micromachines-14-00341-f009:**
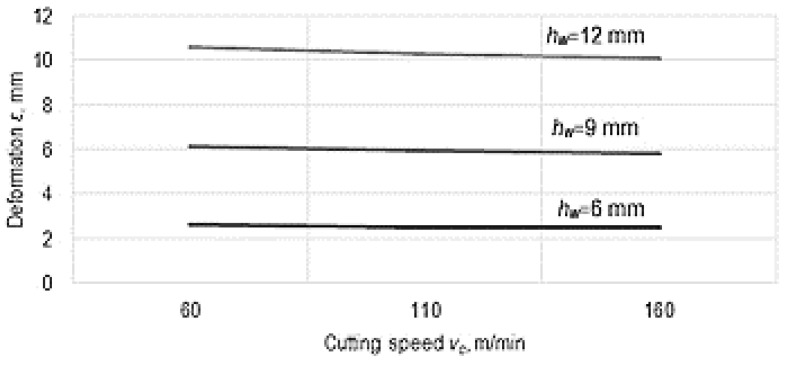
The influence of cutting speed *v_c_* on the deformation ε for three different heights of the thin wall (*f_z_* = 0.0045 mm/tooth, *t_t_* = 1.2 mm, *t_w_* = 0.2 mm, *w_w_* = 10 mm).

**Table 1 micromachines-14-00341-t001:** Conditions and results of experimental tests.

Exp. No.	Input Factors (Coded Value/Real Value)						Wall Deviation, *ε*, mm
	Cutting Speed, Coded Value/, Real Value *v_c_*, m/min/Rotation Speed, *n*, rev/min	Feed Rate, Coded Value/*f_min_*, mm/min/*f_z_*, mm/Tooth	Thickness of the Milling Cutter, Coded Value/Real Value, *t_t_* mm	Wall Thickness, Coded Value/Real Value, *t_w_*, mm	Wall Width, *w_w_*, mm	Wall Height, *h_w_*, mm	
1	1/98.91/500	1/100/0.0047	1/1.2	1/0.2	10	14.7	2.45
					10	9	2.17
					10	6	1.92
					5	14.7	1.83
					5	9	2.21
					5	6	2.03
2	1/98.91/500	1/100/0.0047	1/1.2	2/0.8	10	14.7	11.5
					10	9	6.92
					10	6	4.19
					5	14.7	9.92
					5	9	5.05
					5	6	2.16
3	1/62.31/315	2/100/0.0075	2/1.54	1/0.4	10	14.7	7.70
					10	9	4.19
					10	6	2.23
					5	14.7	6.37
					5	9	3.28
					5	6	2.66
4	1/62.31/315	2/160/0.0075	2/1.54	2/0.8	10	14.7	4.81
					10	9	3.62
					10	6	2.65
					5	14.7	6.85
					5	9	2.81
					5	6	2.79
5	2/98.91/500	1/100/0.0047	2/1.54	1/0.4	10	14.7	4.26
					10	9	4.21
					10	6	2.67
					5	14.7	5.46
					5	9	3.77
					5	6	2.68
6	2/98.91/500	1/100/0.0047	2/1.54	2/0.8	10	14.7	3.46
					10	9	2.88
					10	6	2.54
					5	14.7	2.82
					5	9	2.57
					5	6	2.53
7	2/158.25/800	2/250/0.0074	1/1.2	1/0.2	10	14.7	11.5
					10	9	6.64
					10	6	3.84
					5	14.7	9.30
					5	9	4.25
					5	6	2.60
8	2/158.25/800	2/250/0.0074	1/1.2	2/0.8	10	14.7	3.58
					10	9	2.74
					10	6	2.25
					5	14.7	3.48
					5	9	2.65
					5	6	2.08

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
