# Peer review of "Evaluation of Thin Wall Milling Ability Using Disc Cutters"

_micromachines, 2023, doi:10.3390/mi14020341_

Round 1
Reviewer 1 Report
Dear author(s), please find below suggestions that may justify my final evaluation of the reviewed manuscript ‘Evaluation of thin wall milling ability using disc cutters, Manuscript ID: micromachines-2173437.
Generally, the paper's idea is interesting, the topic is up-to-date, but the contribution to the field is not significant.
1. Put statistical findings of your main achievement in the work in the abstract.
2. Introduction needs references as per the text put in various paragraphs.
3. The introduction needs modifications as the current version did not target the published work in micromachining or thin wall machined structures.
4. Remove all small paragraphs and remove discontinuity between the paragraphs.
5. change the position of figure 1
6. line 194” xOyz”
7. line 197, check the force component, is the out-of-plane deflection in the thin section caused by Fy or Fz or both?
8. How the chip separation in the FE model is achieved. This information is missing. Also, it will be good for the reader to have complete knowledge of tested study parameters before going into FE analysis.
9. Details about the mesh convergence and mesh size are missing.
10. Element details and initial condition used in the FE model is not discussed.
11. From the mesh of the tool, it is obvious that the sharpness of the cutting edge is compromised, needs elaboration.
12. Materials model details are also not provided in the current version.
13. The authors have used the FE model only for energies stored in cutting tools and workpiece, It can also be used for force calculation.
14. The experimental part of the work lacked forces analysis, temperature analysis, surface roughness, and tool wear. The main focus is on the deflection of thin structure, it would be better to include all this information to make the work more attractive for the reader and novel.
15. The material elasticity also plays a vital role in the deformation of thin structures which is not targeted in the current version.
Author Response
Please find the answers for the review in the document attached.

Reviewer 2 Report
Many changes to do, pleae read carefully the attached file

Author Response
Please find the answers to the review in the document attached.

Round 2
Reviewer 1 Report
Dear Authors, thanks a lot for incorporating all the suggested comments and it has certainly improved the quality of your work, my decision regarding your paper is accepted in the current form,
Kind Regards
Reviewer 2 Report
Good revision. Disk cutter is a way to make roughing operations in blisk (balde on disk) and IBRs. Perhaps you can also see those papers about gear that can be a good application in future, as it si the case of 5-axis double-flank CNC machining of spiral bevel gears via custom-shaped milling tools—Part I: Modeling and simulation, Precision Engineering 62, 204-212 and the experimental results in Part II Int J Adv Manuf Technol 119, 1647–1658 (2022). https://doi.org/10.1007/s00170-021-08166-0
Roughing with disks and a posterior work with other tools is a good way for gear making.